

# Self-reported voice problems in call center employees during the COVID-19 pandemic: prevalence, risk factors, and occupational conditions

Songul Demir[1], Bilal Sizer[2] and Sehmus Yildiz[3]

[1] Department of Otorhinolaryngology, Selahattin Eyyubi State Hospital, Diyarbakir, Turkey
[2] Department of Otorhinolaryngology, Assoc. Prof. Dr. Bilal Sizer ENT Clinic, Diyarbakir, Turkey
[3] Department of Speech and Language Therapy, Selahattin Eyyubi State Hospital, Diyarbakir, Turkey

## ABSTRACT

**Background:** The COVID-19 pandemic has significantly altered work environments, especially for call center employees (CCEs), who face increased vocal strain due to prolonged speech, stressful conditions, and changes in work dynamics. This study aims to assess the prevalence of self-reported voice problems among CCEs during the pandemic, focusing on the influence of occupational factors, psychosocial stress, and demographic variables on vocal health.

**Methods:** A cross-sectional survey was conducted with 264 CCEs in Turkey between September 2021 and October 2021 during the pandemic. Participants were individuals who had worked in a call center for at least 6 months, were 18 years or older, had no psychiatric diagnoses, and voluntarily consented to participate; those with a history of voice disorders or vocal cord surgery were excluded. Participants completed a demographic questionnaire and the Voice Handicap Index–Short Form (VHI-SF), a validated tool for measuring the impact of voice disorders on quality of life. Descriptive statistics and non-parametric tests were applied to examine the relationships between workers' characteristics, work conditions, and voice disorder scores.

**Results:** The study sample was comprised of 72% women, with a median age of 25 years (IQR: 22–28). The prevalence of voice disorders was found to be 57.6%. Key risk factors for elevated VHI-SF scores included working more than 8 h daily, having over 5 years of experience, experiencing high stress, and exhibiting poor posture. Workers with inadequate knowledge of proper voice usage or those reporting throat irritation had significantly higher VHI-SF scores. Additionally, those who used their voices for more than 5 h per day, had respiratory allergies, or reported poor vocal health, also exhibited higher VHI-SF scores. No significant correlation was observed between VHI-SF scores and gender, education level, smoking habits or hydration.

**Conclusion:** The study highlights the high prevalence of self-reported voice problems among CCEs during the pandemic and identifies several key occupational and psychosocial risk factors. Interventions targeting voice care education, stress management, and ergonomic work conditions may be beneficial in reducing the incidence and severity of voice disorders in this population.

Corresponding author
Songul Demir, s.gule@hotmail.com

# INTRODUCTION

The global labor market has seen significant changes in recent years, with call centers becoming a pivotal component of the service sector. Driven by globalization and digitalization, the industry has expanded rapidly, supported by advancements such as multichannel communication, artificial intelligence (AI), and remote work.

Although precise global data on call center employees (CCEs) is limited, it is estimated that over 20 million people worldwide work in call centers, with this number rising annually. The United States, India, and the Philippines are among the largest markets for call center services. In 2020, the U.S. employed approximately 2.8 million CCEs, while India and the Philippines had 300,000 and 400,000 workers, respectively, in 2015 (*White, 2015*). In Turkey, the call center sector is also growing rapidly, with 135,820 employees reported in 2020, and an estimated 145,000 expected in 2021 (*Hatman & Torun, 2022*). Turkey has become a prominent provider of outsourced call center services.

The shift to remote work during the COVID-19 pandemic has exposed CCEs to a range of psychosocial and operational challenges, including insufficient technical infrastructure, data security concerns, decreased organizational commitment, attention deficits, and a growing sense of isolation (*Kaushik & Guleria, 2020*).

As a key sector within the service industry, CCEs face elevated risks for various work-related health issues, particularly musculoskeletal problems, psychological stress (such as anxiety, depression, and insomnia), and voice disorders (*Nair et al., 2021*; *Jones et al., 2002*). Factors such as prolonged phone conversations, continuous loud speech, and high-stress customer interactions contribute to the frequent occurrence of vocal pathologies among these workers (*Nair et al., 2021*; *Jones et al., 2002*). Additionally, improper posture, lack of vocal hygiene, smoking, shouting, and gastroesophageal reflux have all been linked to an increased risk of developing voice disorders (*Simberg et al., 2000*). Environmental factors, including noise, poor acoustics, dust, temperature, and inadequate breaks, further exacerbate the likelihood of voice issues in CCEs (*Devadas & Rajashekhar, 2013*).

Common voice symptoms in this group include vocal fatigue, hoarseness, throat irritation, dry mouth, voice loss, excessive vocal effort, and frequent throat clearing (*Nair et al., 2021*). These disorders can lead to reduced productivity, increased absenteeism, and a lower quality of life. In addition, vocal problems can affect work performance and hinder customer satisfaction (*Nair et al., 2021*).

Voice disorders significantly impact daily activities, work performance, social interactions, and psychological well-being. Research shows that individuals with voice disorders often experience distress, low self-esteem, and social withdrawal

(*Misono et al., 2014*). For call center workers, whose daily tasks depend heavily on vocal performance, maintaining good voice health is essential.

To assess the impact of voice disorders on daily life, several assessment tools are commonly used, such as the Voice Handicap Index (VHI), Voice Handicap Index-Short Form (VHI-SF), and Voice-Related Quality of Life (V-RQOL) scales. These tools help to identify voice-related difficulties, monitor complaints, and evaluate quality of life. In this study, the Turkish-adapted version of the Voice Handicap Index-Short Form (VHI-SF) was employed (*Kılıç et al., 2008*).

Research has shown a strong correlation between poor voice use habits and high VHI scores, specifically, prolonged and loud speaking, limited voice rest periods, and the lack of proper vocal techniques can lead to higher VHI scores (*Nair et al., 2021*).

Recommendations aimed at improving voice use habits and the work environment can help maintain the voice health of CCEs. For example, voice hygiene education, teaching proper speaking techniques, and scheduling voice rest periods can contribute to the prevention of voice disorders.

This study aims to investigate the prevalence of self-reported voice problems among CCEs during the COVID-19 pandemic and to examine how daily habits, voice use, and working conditions relate to vocal health. The findings may help guide strategies for the prevention and management of voice problems in this occupational group.

## MATERIALS AND METHODS

The study was conducted between September 2021 and October 2021. Ethical approval was obtained from the Diyarbakır Dicle University Faculty of Medicine's ethics committee (Ethics Committee No: 433, Date: 01.09.2021). Written informed consent was obtained from all participants prior to data collection. The approval from the ethics committee included a detailed review of the study protocol and participant recruitment procedures.

This study employed a cross-sectional survey design to assess the prevalence of self-reported voice problems among CCEs and to explore the relationship between vocal health and various occupational and psychosocial factors.

The sample was determined using convenience sampling. A total of 264 individuals known to be employed in call centers were invited to participate in the study. Participant recruitment was carried out through the distribution of an online survey link, which was shared *via* internal communication channels and professional networks of call centers.

Inclusion criteria for participation were: (1) being 18 years or older; (2) employed in a call center for at least six months; (3) no previous psychiatric diagnoses; and (4) voluntary consent to participate. Individuals were excluded if they had a previously diagnosed voice disorder or a history of vocal cord surgery, or if they declined to participate. Each volunteer was required to complete a demographic questionnaire followed by the Voice Handicap Index–Short Form (VHI-SF). The survey was conducted entirely online, ensuring participants could complete it from home, adhering to the safety guidelines in place due to the pandemic.

**Table 1 Survey questions.**

| Question | Options |
|---|---|
| Daily working hours | 0–6 h, 6–8 h, 8 h or more |
| Do you feel stressed in your work environment? | Yes, No |
| On average, how many hours do you actively use your voice during the day? | 0–1 h, 1–3 h, 3–5 h, 5 h or more |
| How would you describe your water consumption habits? | 0–1 liter/day, 1–2 liters/day, 2–3 liters/day, more than 3 liters/day |
| Do you pay attention to your body posture while speaking? | Yes, No |
| Do you have knowledge about correct voice usage? | Yes, No |
| Do you smoke? | Yes, No |
| Since you started working as a call center employee, have you ever experienced any voice problems? | Yes, No |
| Do you experience throat pain or irritation during the day? | Yes, No |
| Do you have a doctor-diagnosed gastroesophageal reflux? | Yes, No |
| Do you have any known allergies related to your respiratory system? | Yes, No |

The raw data from the completed surveys were stored securely and anonymized to maintain participant confidentiality.

## Assessment tools

The assessment survey form was provided to volunteers who met the inclusion criteria (Table 1). An additional eligibility verification survey was used to ensure participants met the inclusion criteria, such as age, call center experience, absence of psychiatric conditions, and willingness to participate. This step helped confirm that only eligible individuals were included in the study. The results were collected in the researcher's digital data repository.

## Voice handicap index-short form Turkish adaptation version

In studies focusing on voice disorders among CCEs, various questionnaires and scales are typically used to assess voice problems. These types of surveys are designed to objectively evaluate workers' voice health and monitor symptoms. The VHI is one of the most commonly used questionnaires to assess the impact of voice disorders on quality of life and is frequently used among CCEs. The VHI measures the personal, social, and functional effects of voice disorders and helps to understand the impact of voice problems on individuals' daily lives (*Rechenberg, de Goulart & Roithmann, 2011*; *Zhao et al., 2020*). The original test consists of 30 questions, and the abbreviated form (VHI-SF) contains 10 questions. The VHI-SF is a widely used test that has been translated into many languages and employed in various studies (*Sizer, Demir & Atay, 2022*; *Tafiadis et al., 2020*). This short form is a more concise version of the VHI and consists of 10 items (Table 2). The VHI-SF aims to provide a quicker and more practical evaluation of the severity of voice disorders and their impact on individuals' quality of life. Studies on the validity and reliability of the Turkish versions of the VHI and VHI-SF have shown that these adapted versions produce effective and reliable results in the Turkish-speaking population as well (*Kılıç et al., 2008*).

**Table 2  VHI-10 statement-original form.**

| | |
|---|---|
| 1. | My voice makes it difficult for people to hear me. 0 1 2 3 4 |
| 2. | People have difficulty understanding me in a noisy room. 0 1 2 3 4 |
| 3. | People ask, "What's wrong with your voice?" 0 1 2 3 4 |
| 4. | I feel as though I have to strain to produce my voice. 0 1 2 3 4 |
| 5. | My voice difficulties restrict my personal and social life. 0 1 2 3 4 |
| 6. | The clarity of my voice is unpredictable. 0 1 2 3 4 |
| 7. | I feel left out of conversations because of my voice. 0 1 2 3 4 |
| 8. | My voice problem causes me to lose income. 0 1 2 3 4 |
| 9. | My voice problem upsets me. 0 1 2 3 4 |
| 10. | My voice makes me feel handicapped. 0 1 2 3 4 |

Note:
0–Never, 1–Almost Never, 2–Sometimes, 3–Almost Always, 4–Always.

The VHI-SF consists of 10 questions that evaluate the physical, social, and psychological impacts of voice disorders. Each question is scored on a scale from 0 to 4:

0: No problem,
1: Occasionally mild problems,
2: Moderate problems,
3: Severe problems,
4: Constant and severe problems

When the scores are summed, the total score reflects the severity of the voice disorder and its impact on the individual's quality of life. Higher scores indicate greater effects. Considering the prevailing Covid-19 pandemic conditions during the study period, the survey was administered online.

## Statistical analysis

Descriptive statistics were used to summarize the demographic data, and non-parametric statistical tests were applied to assess the relationships between participant characteristics, work conditions, and voice disorder scores. Descriptive data were presented as frequencies and percentages, while numerical data were represented using the median and interquartile range (IQR) as appropriate. The normality assumption of the numerical data was tested using the Kolmogorov-Smirnov test and histograms. The Mann-Whitney U test and Kruskal-Wallis test were applied to compare VHI scores between groups where appropriate. Spearman correlation analysis was used to examine the correlation between two measurements. A $p$-value of <0.05 was considered statistically significant. Bonferroni correction was applied for *post hoc* analyses. All analyses were performed using IBM SPSS Statistics 20 (IBM Corp., Armonk, NY, USA).

## RESULTS

A total of 72% of the participants ($n$ = 190) were female, and 28% ($n$ = 74) were male. The median age was 25 (IQR: 22-28). Regarding educational status, 74.2% ($n$ = 196) were

**Table 3 Distribution of participants according to survey parameters.**

| | | n (%) |
|---|---|---|
| Gender | Female | 190 (72.0) |
| | Male | 74 (28.0) |
| Age | | 25 (22–28) |
| Education level | High school | 68 (25.8) |
| | University | 196 (74.2) |
| Years of experience | 0–1 year | 190 (72.0) |
| | 1–3 year | 36 (13.6) |
| | 3–5 year | 20 (7.6) |
| | 5+ years | 18 (6.8) |
| Daily working hours | 6–8 h | 236 (89.4) |
| | 8+ h | 28 (10.6) |
| Do you feel stressed in your working environment? | No | 110 (41.7) |
| | Yes | 154 (58.3) |
| How many hours do you actively use your voice daily? | 1–3 h | 10 (3.8) |
| | 3–5 h | 234 (88.6) |
| | 5+ h | 20 (7.6) |
| How would you describe your water intake habits? | 0–1 liter | 88 (33.3) |
| | 1–2 liters | 92 (34.8) |
| | 2–3 liters | 58 (22.0) |
| | 3+ liters | 26 (9.8) |
| Do you pay attention to your posture while speaking? | No | 86 (32.6) |
| | Yes | 178 (67.4) |
| Do you have knowledge of proper voice use? | No | 98 (37.1) |
| | Yes | 166 (62.9) |
| Do you smoke? | No | 170 (64.4) |
| | Yes | 94 (35.6) |
| Have you had any voice problems since you started working as a call center employee? | No | 112 (42.4) |
| | Yes | 152 (57.6) |
| Do you experience throat pain or irritation during the day? | No | 72 (27.3) |
| | Yes | 192 (72.7) |
| Do you have a diagnosed gastroesophageal reflux disease (GERD)? | No | 232 (87.9) |
| | Yes | 32 (12.1) |
| Do you have known respiratory allergies? | No | 224 (84.8) |
| | Yes | 40 (15.2) |
| How is your voice today? | Normal | 158 (59.8) |
| | Mildly impaired | 62 (23.5) |
| | Moderately impaired | 44 (16.7) |
| How would you describe your speaking voice use? | I speak very little | 22 (8.3) |
| | I speak normally | 158 (59.8) |
| | I speak a lot | 84 (31.8) |
| How would you describe your singing voice use? | I never sing | 60 (22.7) |
| | I sing occasionally | 166 (62.9) |
| | I sing very frequently | 38 (14.4) |

**Table 4 Comparison of participants' VHI scores based on study parameters.**

| | | VHI score Median IQR | p |
|---|---|---|---|
| Gender | Female | 2.0 (0.0–7.0) | 0.125[m] |
| | Male | 3.0 (1.0–8.0) | |
| Education level | High school | 2.5 (0.0–8.0) | 0.640[m] |
| | University | 3.0 (0.0–7.0) | |
| Years of experience | 0–1 year[1] | 3.0 (0.0–6.0) | **0.011**[k] |
| | 1–3 years[2] | 2.5 (0.0–9.0) | |
| | 3–5 years[3] | 0.0 (0.0–7.0) | |
| | 5+ years[4] | 10.5 (1.0–20.0) | |
| Daily working hours | 6–8 h | 2.0 (0.0–7.0) | **0.029**[m] |
| | 8+ h | 4.5 (2.0–8.0) | |
| Do you feel stressed in your working environment? | No | 1.0 (0.0–4.0) | **<0.001**[m] |
| | Yes | 5.0 (1.0–9.0) | |
| Daily hours of active voice use | 1–3 h[1] | 4.0 (0.0–7.0) | **0.005**[k] |
| | 3–5 h[2] | 2.0 (0.0–7.0) | |
| | 5+ h[3] | 5.0 (4.0–12.0) | |
| How would you describe your water intake habits? | 0–1 litre | 2.0 (0.0–6.0) | 0.454[k] |
| | 1–2 litre | 3.0 (0.0–8.0) | |
| | 2–3 litre | 2.0 (0.0–7.0) | |
| | 3 litre ve daha fazla | 3.0 (1.0–4.0) | |
| Do you pay attention to your posture while speaking? | No | 4.0 (1.0–9.0) | **0.010**[m] |
| | Yes | 2.0 (0.0–7.0) | |
| Do you have knowledge of proper voice use? | No | 4.0 (1.0–8.0) | **0.003**[m] |
| | Yes | 2.0 (0.0–6.0) | |
| Do you smoke? | No | 3.0 (0.0–6.0) | 0.326[m] |
| | Yes | 3.0 (1.0–8.0) | |
| Have you had any voice problems since you started working as a call center employee? | No | 1.0 (0.0–3.0) | **<0.001**[m] |
| | Yes | 5.0 (1.0–9.0) | |
| Do you experience throat pain or irritation during the day? | No | 1.0 (0.0–2.0) | **<0.001**[m] |
| | Yes | 4.0 (1.0–8.5) | |
| Do you have a diagnosed gastroesophageal reflux disease (GERD)? | No | 3.0 (0.0–7.0) | 0.768[m] |
| | Yes | 1.0 (0.0–11.0) | |
| Do you have known respiratory allergies? | No | 2.5 (0.0–7.0) | **0.032**[m] |
| | Yes | 6.0 (1.0–8.5) | |
| How is your voice today? | Normal | 1.0 (0.0–4.0) | **<0.001**[k] |
| | Mildly impaired | 6.0 (2.0–8.0) | |
| | Moderately impaired | 7.5 (3.0–13.0) | |
| How would you describe your speaking voice use? | I speak very little | 4.0 (0.0–7.0) | 0.947[k] |
| | I speak normally | 3.0 (0.0–8.0) | |
| | I speak a lot | 2.0 (0.0–7.0) | |
| How would you describe your singing voice use? | I never sing | 5.0 (2.0–9.0) | **0.018**[k] |
| | I sing occasionally | 2.0 (0.0–7.0) | |
| | I sing very frequently | 2.0 (0.0–8.0) | |

**Notes:**
[m] Mann Whitney U test.
[k] Kruskal Wallis test.
*Post hoc* adjusted p values for; Years of Experience: 1–2: $p = 0.999$, 1–3: $p = 0.999$, 1–4: $p = 0.009$, 2–3: $p = 0.999$, 2–4: $p = 0.047$, 3–4: $p = 0.026$.
Daily Hours of Active Voice Use: 1–2: $p = 0.999$, 1–3: $p = 0.177$, 2–3: $p = 0.003$. Bold values indicate statistically significant differences between the groups ($p < 0.05$).
university graduates, while 25.8% ($n = 68$) were high school graduates. In terms of work experience, 72% of the participants ($n = 190$) had 0–1 years of experience, and 10.6% ($n = 28$) worked more than 8 h a day (Table 3).

In the work environment, 58.3% of the participants ($n = 154$) reported feeling stressed, and 7.6% ($n = 20$) actively used their voices for more than 5 h daily. In terms of water consumption habits, 34.8% ($n = 92$) consumed 1–2 liters of water per day, 22% ($n = 58$) consumed 2–3 liters, and 9.8% ($n = 26$) consumed 3 liters or more. A total of 67.4% ($n = 178$) of the participants paid attention to their posture while speaking, and 62.9% ($n = 166$) were knowledgeable about correct voice use (Table 3).

After starting their job as a call center employee, 57.6% ($n = 152$) of the participants reported experiencing voice problems. Regarding their current voice condition, 59.8% ($n = 158$) stated that their voices were not impaired, while 31.8% ($n = 84$) reported speaking excessively. In terms of singing habits, 62.9% ($n = 166$) occasionally sang, while 14.4% ($n = 38$) sang very frequently (Table 3).

The table shows that the majority of the participants had work habits consistent with their job demands and voice use, and they paid attention to their voice health (Table 3).

## Comparison of VHI scores according to research parameters

When comparing participants' VHI scores based on research parameters, those with 5 or more years of experience had significantly higher VHI scores compared to other groups ($p = 0.011$). Additionally, participants working more than 8 h daily had higher scores than those working 6–8 h ($p = 0.029$). VHI scores were significantly higher in participants who reported feeling stressed ($p < 0.001$), those who did not pay attention to posture ($p = 0.010$), those who lacked knowledge about correct voice use ($p = 0.003$), and those with complaints of sore throat or irritation ($p < 0.001$), compared to participants without these characteristics (Table 4).

Participants who used their voices actively for more than 5 h daily had higher VHI scores compared to those who used their voices for 3 to 5 h daily ($p = 0.005$). Furthermore, individuals with respiratory allergies ($p = 0.032$), those with poor voice condition ($p < 0.001$), and those who never sang ($p = 0.018$) had higher VHI scores compared to participants without these characteristics. Additionally, participants who reported their voice as "not impaired" had significantly lower VHI scores compared to those who described their voice as "slightly impaired" or "moderately impaired" (Table 4).

In contrast, no significant relationship was found between VHI scores and variables such as gender, educational status, water consumption habits, smoking status, or speech volume use (Table 4). Moreover, no significant correlation was found between age and VHI scores ($p = 0.485$).

## DISCUSSION

The findings from this study indicate that CCEs are at high risk for voice health issues, and factors such as working conditions, voice usage habits, and psychosocial stress significantly exacerbate these risks. These results are largely consistent with existing literature,

highlighting the severity of voice-related issues faced by CCEs (*Nair et al., 2021*; *Jones et al., 2002*; *Devadas & Rajashekhar, 2013*).

*Nair et al. (2021)* in their systematic review, including 15 studies, reported that the prevalence of voice problems among CCEs ranged from 33% to 68%. *da Silva Dantas et al. (2023)* in a systematic review of literature on voice disorders among CCEs published in 2023, stated that the prevalence of voice problems among CCEs typically ranges from 30% to 60%. In this study, the prevalence of voice disorders among CCEs was found to be 57.6%, which is consistent with the literature.

In this study, a large proportion of participants reported voice usage habits that align with their work intensity and mentioned that they pay attention to voice health.

Long hours of phone conversations are an important risk factor for voice disorders. As daily working hours increase, the prevalence of voice problems also rises. This is mainly due to insufficient attention to voice hygiene, lack of adequate rest, and constant use of the voice at high volumes. Studies have shown that sustained voice use, especially for 6 h or more a day, increases symptoms such as hoarseness, fatigue, and sore throat (*Nair et al., 2021*; *Jones et al., 2002*). In this study, participants who worked 8 or more hours daily had significantly higher VHI scores, indicating a positive relationship between daily working hours and voice disorders. Based on these findings, it can be recommended that employers regulate the daily working hours of CCEs to help preserve voice health. Additionally, providing voice hygiene training and voice rest breaks may help prevent voice disorders.

*Hatman & Torun (2022)* demonstrated that among CCEs requiring vocal rehabilitation, prolonged occupational exposure was significantly associated with an increased risk of developing voice disorders. These findings align with existing literature emphasizing the cumulative impact of vocal load and occupational stress on vocal health among professional voice users. This study also highlights the increased risk of voice disorders due to more vocal strain as employees accumulate more years of experience. In this study, CCEs with 5 or more years of experience had higher VHI scores. However, most of the volunteers in our study had between 0 and 1 year of experience. Upon further investigation, we found that the majority of workers leave the job after one year due to high work pressure and stress. Therefore, the heterogeneity of groups in terms of work duration can be attributed to this factor. Additionally, no significant correlation was found between age and VHI scores in our study. However, *Jones et al. (2002)* considered age and experience as other important demographic factors affecting voice health in their study. According to *Jones et al. (2002)* young and inexperienced workers may pay less attention to their voice health, which may lead to more voice problems.

In this study, VHI scores were higher in participants who reported feeling stressed, who did not pay attention to posture, and who lacked knowledge about proper voice use. Specifically, job-related stress is directly linked to voice problems. CCEs face psychological pressures due to high work pace and customer demands, which may negatively impact their voice health. *Devadas & Rajashekhar (2013)* suggested that stress management and psychosocial support programs for CCEs could be beneficial in reducing voice problems and improving the overall health of workers.

Long working hours alone are not the only risk factor; they are compounded by poor voice hygiene (*e.g.*, insufficient water intake, vocal strain), ergonomic issues (*e.g.*, incorrect microphone usage or improper body posture), and can lead to an increased frequency of voice disorders. In this context, a study investigating the relationship between voice disorders and lack of voice hygiene among CCEs found a strong association between poor voice hygiene habits and the emergence of voice disorders (*Fuentes-López, Fuente & Contreras, 2019*). This study also noted that CCEs often work in ergonomically unfavorable positions, which could negatively impact their voice health. Incorrect microphone usage, excessive communication with the voice, and uncomfortable working environments emerged as major threats to voice health (*Fuentes-López, Fuente & Contreras, 2019*). These findings underscore the importance of preventive measures such as maintaining voice hygiene, implementing ergonomic adjustments, and providing voice training in the workplace.

Several studies have examined the impact of demographic factors on voice health. *Jones et al. (2002)* noted that female CCEs tend to experience more voice problems than male workers. The researchers attributed this to the physiological differences in female vocal cords and the voice usage habits of women, which might affect their voice health more significantly. In another study, *Lehto et al. (2005)* examined the voice symptoms of call center workers and observed that women experience more voice symptoms compared to men. The researchers concluded that women use their voices more intensively and, therefore, greater attention should be given to their vocal health (*Lehto et al., 2005*). In the study by *Devadas & Rajashekhar (2013)*, the prevalence of vocal symptoms such as hoarseness and vocal fatigue was found to be similar between male and female CCEs. However, it was observed that women were more likely to recognize these symptoms, such as vocal fatigue, as signs of illness and paid more attention to their health compared to men (*Devadas & Rajashekhar, 2013*). In this study, no significant increase in VHI scores was found based on gender.

This study found that CCEs with respiratory allergies had higher VHI scores compared to those without this condition. *Andrea et al. (2024)* in their study with professional singers, found that VHI scores were significantly higher in the group with allergic rhinitis. The authors emphasized that treatment for allergic rhinitis is an important component of professional voice health, especially for preventing vocal strain and subsequent functional or organic laryngeal pathologies. We believe that similar recommendations should be applied to CCEs with respiratory allergies.

In this study, there was no significant increase in VHI scores among participants who were smokers. According to the study by *Hatman & Torun (2022)*, smoking rates were higher among call center operators diagnosed with voice-related occupational diseases, although the difference between those diagnosed and those not diagnosed was not statistically significant.

This study did not find a significant relationship between education level and VHI scores. However, in a study by *Kim et al. (2016)* evaluating individuals with voice disorders according to their sociodemographic characteristics, it was emphasized that individuals with higher education levels had a lower prevalence of voice disorders. The authors

suggested this may be due to their greater attention and care for their voice (*Kim et al., 2016*).

Hydration is strongly linked to voice disorders, and proper hydration has a positive effect on voice health. Studies have shown that insufficient hydration negatively affects vocal quality by reducing the efficiency and elasticity of the vocal folds, thereby increasing the risk of voice disorders (*Pilsl et al., 2025*). The findings of *Pilsl et al. (2025)* support the implementation of hydration strategies, including superficial methods such as lozenges or steam inhalation, as preventive measures for individuals in vocally demanding occupations. However, no significant relationship was found between hydration habits and VHI scores in this study.

The COVID-19 pandemic has introduced numerous new factors that negatively affect the psychological and physical health of CCEs. According to *Santiago et al. (2021)* during the pandemic, CCEs experienced increased stress levels in both their work and personal lives, which led to burnout.

*Sarımehmetoğlu & Barmak (2024)* emphasize the need for a more detailed study on voice risk factors and preventative strategies as call center companies expand globally and the number of employees increases. General observations during this study suggest that changes in the working environment, increased workload, and extended working hours during the pandemic, along with the shift to remote work, exacerbated vocal fatigue and negatively impacted vocal health. Based on these findings, we recommend specific strategies to improve the vocal health of call center workers, such as optimizing work conditions, incorporating regular vocal breaks, and providing training on proper voice use and self-care practices.

### Limitations

This study has several limitations that should be considered when interpreting the results. First, the cross-sectional design limits the ability to establish causal relationships between occupational factors and voice disorders. Longitudinal studies are needed to better understand the long-term effects of work conditions on vocal health. Second, the sample was limited to CCEs in Turkey, which may limit the generalizability of the findings to other regions or professions. Additionally, the use of self-reported data may introduce recall bias or inaccuracies in reporting voice problems. Lastly, although various potential risk factors were examined, other factors, such as other environmental conditions or genetic predisposition, were not considered in this study. Future research could expand on these limitations to provide a more comprehensive understanding of voice disorders among CCEs.

### CONCLUSION

This study reveals a high prevalence of self-reported voice problems among CCEs during the COVID-19 pandemic, with 57.6% reporting voice disorders. Key risk factors include prolonged phone use, long working hours, stress, poor posture, and lack of voice care knowledge. CCEs with over 5 years of experience, high stress, and throat irritation had significantly higher VHI scores.

The findings highlight the importance of addressing work conditions and stress to improve vocal health. Interventions such as voice hygiene education, ergonomic adjustments, and stress management programs are crucial. Employers should prioritize vocal health and provide proper training to reduce voice-related issues in this workforce.

## ACKNOWLEDGEMENTS

The authors would like to express sincere gratitude to their family for their unwavering support and encouragement throughout the course of this study.

### Funding
The authors received no funding for this work.

### Competing Interests
The authors declare that they have no competing interests.

### Author Contributions
- Songul Demir conceived and designed the experiments, performed the experiments, analyzed the data, prepared figures and/or tables, authored or reviewed drafts of the article, and approved the final draft.
- Bilal Sizer conceived and designed the experiments, performed the experiments, analyzed the data, prepared figures and/or tables, authored or reviewed drafts of the article, and approved the final draft.
- Sehmus Yildiz conceived and designed the experiments, performed the experiments, authored or reviewed drafts of the article, and approved the final draft.

### Human Ethics
The following information was supplied relating to ethical approvals (*i.e.*, approving body and any reference numbers):

The study was approved by the Ethics Committee of Dicle University Faculty of Medicine, approval number 2021/433, dated 01.09.2021.

### Data Availability
The raw data is available in the Supplemental File.

### Supplemental Information
Supplemental information for this article can be found online at http://dx.doi.org/10.7717/peerj.19595#supplemental-information.

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
