# Peer review of "Self-reported voice problems in call center employees during the COVID-19 pandemic: prevalence, risk factors, and occupational conditions"

_PeerJ, doi:10.7717/peerj.19595_

## Round 0.1 · original submission · Major Revisions

Please address all the reviewer comments.

**Language Note:** The review process has identified that the English language must be improved. PeerJ can provide language editing services - please contact us at [email protected] for pricing (be sure to provide your manuscript number and title). Alternatively, you should make your own arrangements to improve the language quality and provide details in your response letter. – PeerJ Staff

·

Basic reporting

Thank you for giving me the opportunity to review the manuscript titled “Voice Disorders in Call Center Workers During the COVID-19 Pandemic: Prevalence, Risk Factors, and Occupational Conditions.” This manuscript highlights the increased risk of occupational voice problems among call centre workers and aims to establish the prevalence of these issues during the COVID-19 pandemic. Additionally, the authors seek to identify various risk factors that contribute to voice problems within this population.

Major comments:
1. The reporting style used in the manuscript is adequate. The authors have successfully highlighted the importance of selecting the population of interest and have reviewed existing literature on voice problems in call center workers. However, there is a major concern regarding the reporting of content in the manuscript. Although the title says, “Voice disorders in call center workers during the COVID-19 pandemic,” the introduction section of the manuscript does not report how or why this study is important during the COVID-19 pandemic. This information is essential as it helps readers understand the specific challenges faced by call center workers during the pandemic, such as increased vocal load leading to voice problems. If this information is added, it would justify the specific purpose of the study.
2. The discussion section needs some reorganization and trimming to include only necessary information. While the authors have thoroughly discussed several findings, it is essential to ensure that statements are clear and concise to avoid confusion. For example, the statement in lines 245-246, "In this study, the validity and reliability analyses of the test were conducted, and the Turkish-adapted version was used (9)," is unclear. It is not evident whether the authors are referring to the established validity and reliability of the VHI-SF or to an examination of validity and reliability in the present study (which has not been conducted). Such statements need to be clarified by the authors to prevent misunderstandings.
3. Like the first comment, the authors need to discuss as to how their findings from the survey and administration of VHI-SF are valid in the context of COVID-19 pandemic to the population studies. This information is missing currently from the manuscript (except in lines 329-330) and needs to be added.

Minor comments:
1. The authors have used the term “Call Center Employees (CCEs)” in the introduction section (lines 72, 75, 105, 108), while in the abstract and the discussion section, they have used “Call Center Workers (CCW)” (lines 45, 49, 63, 229, 234, 235, 241, and so on). It would be helpful if the authors consistently used the same abbreviation when referring to the same population throughout the manuscript.
2. The authors need to provide literature support for statements made in the introduction section (e.g., lines 81-83, 99, 100, 101, etc.). Similarly, some statements in the discussion section lack citations, even though they are attributed to previous findings or existing literature (e.g., lines 233, 290-292, 322-323). Adding appropriate references will enhance the credibility and reliability of the manuscript.
3. The statement in lines 244-245, “The VHI is a widely used test that has been translated into many languages and employed in various studies (12,13),” can be modified to focus on the 10-item short version of VHI, which is used in this study. Several recent studies have translated and adapted the VHI to multiple languages. Citing these studies would enhance the literature support for the authors’ statement.
4. While reporting the methods within the abstract, (lines 49-50), the time frame of data collection could be included. This would help a reader better understand the context of this study.

Experimental design

The research carried out as part of this study adheres to the scope of this journal. The research design used is sufficient and effectively addresses the research questions posed. The authors have mentioned that the study has received approval from the institution's Ethics Committee and that they have followed ethical procedures during the analysis (lines 139-140). However, the following clarifications are sought from the authors:

Major comments:
1. It would be beneficial to explicitly state the aim and objectives of the study towards the end of the introduction section. This will help readers better understand the purpose and importance of the study.
2. Adding information on how the participants were identified would help bring more clarity to the recruitment process. Additionally, explicitly stating the sampling method used would enhance the manuscript's understanding for the scientific community.
3. The authors have stated, “This survey was distributed to participants using an online link, considering the Covid-19 pandemic conditions that prevailed during the study period” (lines 127-128). It would be beneficial to mention the specific platform used to conduct the survey in the manuscript for added clarity.
4. Could the authors provide more information on how they ensured that the "264 individuals who were known to be employed in call centers" (lines 126-127) met the inclusionary and exclusionary criteria of this study (lines 129-133)?
5. In continuation to the previous comment, it would be helpful to document the response rate of this survey (i.e., out of the total number of prospective participants, how many responded). This is an essential metric for reporting survey studies (Sharma et al., 2021) and has been documented in almost all previous studies in this field.
6. The authors can consider reporting the findings of the post hoc analysis carried out after the Kruskal-Wallis test. This will help readers better understand the context of specific factors being significantly different. For example, in years of experience, there is a significant difference between 0-1 year, 1-3 years, 3-5 years, and 5+ years of experience (p = 0.011). Similarly, in hours of active voice use, there is a significant difference between 1-3 hours, 3-5 hours, and 5+ hours (p = 0.005). Readers would be interested in knowing which groups were significantly different from each other. Documenting the findings from the post hoc analysis can effectively convey this information and help the authors better discuss and reason out their findings.
7. It is unclear where the authors included the last three questions mentioned in Table 3 ("How is your voice today?", "How would you describe your speaking voice use?", and "How would you describe your singing voice use?"). These questions are not found in Table 1 (survey questions) or Table 2 (VHI-10). When encountered suddenly, this might confuse the reader. Please clarify this information to make the methodology clearer.

Minor comments:
1. It might be helpful to avoid repetition within the methods section. For example, there are three instances mentioning that the study was conducted online due to the COVID-19 pandemic (lines 127-128; 145-146; 163-165).
2. The sentence “Descriptive statistics were used to summarize the … work conditions, and voice disorder scores.” (lines 140-142) could be merged with the section on statistical analysis. This would make the content appear more coherent.
3. In Table 4, "daily working hours" is indicated twice in the first column. This could be confusing for readers. It would be helpful to change the second instance to "active voice use" or "hours of active voice use," as mentioned in the questionnaire (Q.3 in the survey questions in Table 2).
4. In Table 4, while comparing the responses to the question “Do you feel stressed in your working environment?” using the Mann-Whitney U test, the p-value is significant (p <0.001). This should be indicated in bold to clearly indicate its significance.

Validity of the findings

Major comments:
1. The term "voice disorders" might not be entirely appropriate in the context of this research. Instead, the authors could consider using "voice problems" or more specifically "self-reported voice problems." This is because the study is based on self-reports from participants (survey questionnaire and VHI-SF), and no objective assessment has been conducted by professionals to determine whether the individuals in this study (i.e., the call center workers) have a voice disorder. Therefore, it is advisable that the authors change the term "voice disorders" to "voice problems" or "self-reported voice problems" throughout the manuscript, including the abstract and title. This adjustment will help ensure the validity of the findings.
2. In line 226, the authors state, “Moreover, no significant correlation was found between age and VHI scores (Table 4).” However, this information is not presented in Table 4. It would be helpful to add the correlation information either within Table 4 or as a running text towards the end of the results section. Even though the correlation is not statistically significant, providing the p-value and reporting the correlation coefficient would enhance the validity of your results.
3. In lines 295-301, the authors discuss how demographic factors impact vocal health, citing references 4 and 17 to support their findings. However, reference 17 may not be entirely relevant to the current context, as it focuses on teachers. Instead, the authors could refer to the study by Lehto et al., 2005, which reports a gender effect among call center workers. Additionally, the study by Gupta & Sekher (2023) reviews general health implications for female call center workers. These studies could be cited. The findings of Devadas & Rajashekar (2013), who did not find a gender effect like the present findings can also be added to strengthen any possible explanation as to why you did not find gender effect on voice health.
4. In lines 308-313, the authors discuss the lack of a significant increase in VHI scores among participants who were smokers. However, the cited literature from Ben-David & Icht (2016) is based on objective voice parameters and does not involve the self-rating measure of VHI. Therefore, it would be more appropriate to find literature that specifically addresses the self-reported VHI scores in relation to smoking. This would strengthen the validity of your findings.
5. Throughout the discussion section or towards the end, it would be beneficial for the authors to suggest changes to improve the vocal health of call center workers in relevance to each finding of the present study. This will assist clinicians in applying the findings from this research into clinical practice. Although there is currently a statement in lines 328-330, refining it to focus specifically on vocal health, which is presumed to be the aim of the study, would enhance the manuscript.
6. It would be helpful if the authors could add a section on the limitations of their study towards the end of the discussion section. This would provide readers with a more comprehensive understanding of the study's scope and any potential constraints.
7. Conclusions of the study, presently focusses on the implications. However, the authors can consider rewriting the conclusion section to focus only on the important findings from the present study and its implications.

Additional comments

References:

Devadas, U., & Rajashekhar, B. (2013). The prevalence and impact of voice problems in call center operators. Journal of Laryngology and Voice, 3(1), 3-9.
Gupta, A., & Sekher, T. V. (2023). Call Centers and Associated Health Hazard for Women Employees: A Review of Health Implications for Women Employees of Transnational Call Centers in India. SAGE Open, 13(3), 21582440231192152.
Lehto, L., Alku, P., Bäckström, T., & Vilkman, E. (2005). Voice symptoms of call-centre customer service advisers experienced during a work-day and effects of a short vocal training course. Logopedics phoniatrics vocology, 30(1), 14-27.
Sharma, A., Minh Duc, N. T., Luu Lam Thang, T., Nam, N. H., Ng, S. J., Abbas, K. S., ... & Karamouzian, M. (2021). A consensus-based checklist for reporting of survey studies (CROSS). Journal of general internal medicine, 36(10), 3179-3187.

·

Basic reporting

Abstract:
Please add a sample size justification.
Mention Inclusive and exclusive criteria.
Introduction:
The introduction provides a strong foundation, clearly explaining the importance of the research topic.
It would be helpful to include more recent studies (2023–2024) to strengthen the justification.
The knowledge gap being addressed is well-stated but could be highlighted more explicitly.
Materials and Methods:
Add Sample size justification.
Mention Sampling technique.
Results:
You mentioned that the age is 18 or above, and in the results section, you mentioned 25(22-28). What is this number of %?
Discussion:
Please remove the 1st and 2nd paragraph for discussion, it may be part of materials and methods.
Please do not use words like our, my, we, or me.
It is better not to use a review article in a discussion.
Conclusion:
Please rewrite the conclusion, and that should match the conclusion with the title and objective.

Experimental design

This is cross sectional survey.

Validity of the findings

Abstract:
Please add a sample size justification.
Mention Inclusive and exclusive criteria.
Introduction:
The introduction provides a strong foundation, clearly explaining the importance of the research topic.
It would be helpful to include more recent studies (2023–2024) to strengthen the justification.
The knowledge gap being addressed is well-stated but could be highlighted more explicitly.
Materials and Methods:
Add Sample size justification.
Mention Sampling technique.
Results:
You mentioned that the age is 18 or above, and in the results section, you mentioned 25(22-28). What is this number of %?
Discussion:
Please remove the 1st and 2nd paragraph for discussion, it may be part of materials and methods.
Please do not use words like our, my, we, or me.
It is better not to use a review article in a discussion.
Conclusion:
Please rewrite the conclusion, and that should match the conclusion with the title and objective.

Additional comments

please review English. Published after the revision.

---

## Round 0.2 · accepted · Accept

The authors responded to the comments raised by the reviewers appropriately.

·

Basic reporting

No comments

Experimental design

No comments

Validity of the findings

No comments

Additional comments

Thank you for giving me the opportunity to review the revisions made by the authors.

The authors have incorporated the suggested modifications and made significant improvements to the manuscript. I have no additional remarks, and the manuscript can be accepted.

·

Basic reporting

All suggested changes were included in the manuscript.

Experimental design

All suggested changes were included in the manuscript.

Validity of the findings

All suggested changes were included in the manuscript.

Additional comments

The article meets the PeerJ criteria and should be published.